# Social Cohesion and Community Resilience during the COVID-19 Pandemic in Northern Romania

**DOI:** 10.3390/ijerph19084587

**Published:** 2022-04-11

**Authors:** Despina Saghin, Maria-Magdalena Lupchian, Daniel Lucheș

**Affiliations:** 1Department of Geography, Stefan cel Mare University of Suceava, 720229 Suceava, Romania; mmlupchian@atlas.usv.ro; 2Department of Sociology, West University of Timișoara, 300223 Timișoara, Romania; daniel.luches@e-uvt.ro

**Keywords:** COVID-19, civil society, mass media, individual and collective resilience

## Abstract

The COVID-19 pandemic and the lock-down have highlighted the growing awareness of the need to involve the population in solving problems that directly affect the existence and trajectory of the life of the individual and civil society in the local, national, and regional context. The article aims both to analyze the reaction of formal and informal civil society in a context of major crisis and to analyze how the population perceives the involvement of civil society at the level of a county in Romania and its county seat city. The present sociological diagnosis used data that were collected through an online survey at the beginning of May 2020 among the population of Suceava County. It was sought to identify how the reaction of civil society was perceived and how it was mobilized, as well as how the mass media contributed to reducing the effects of the pandemic. After the elimination phase of non-compliant responses, the volume of the sample included a total of 1231 people. The results of the study indicate that this pandemic context, which manifested as a major crisis, also had positive effects in the sense of mobilizing latent but extensive energies at the local level, whose manifestation contributed to diminishing and limiting the effects of the sanitary crisis the county faced. The media, as a component of civil society, has managed to mobilize important segments of the population, both in quarantined localities and in other localities in Suceava County and Moldova. The COVID-19 crisis tested the social cohesion and resilience of communities and offered perhaps one of the most remarkable lessons of solidarity in the post-December period, both locally and nationally. Although the perception of Romanians on the role of civil society would rather be part of a culture of individualism, in extreme situations it was found that its activity has never been more important.

## 1. Introduction

The pandemic that has marked and continues to mark humanity since 2020 has had a strong impact on the health of the population, on health systems around the world, and on economic and social life. At the same time, the architecture of social relations has changed, both as a result of restrictions imposed by the authorities and as a result of personal decisions about the need to protect against an enemy that is still little known. Under these conditions, values such as solidarity with the most vulnerable to the disease, concrete involvement in helping them, and responsibility towards others and towards the community to which you belong have acquired new forms of manifestation and meanings. The COVID-19 crisis is proving to be one of the most significant paradigm shifts in the contemporary world.

Along with studies analyzing the impact of the pandemic on health, education, and economic systems both globally and locally, concerns have been raised about how society has reacted to this major crisis [1,2,3,4,5]. This shapes the idea that the current context has led to an increase in creativity in the manifestation of solidarity [6,7], and this boost in creativity was all the greater as the experience in showing solidarity was less.

Resilience is a social process through which local communities implement collective and individual actions in order to reduce the negative effects of threats through an effective response and adaptation. It has been the subject of numerous studies (Community and Regional Resilience Institute [8,9,10,11,12]). Major crises are accompanied or followed by many negative consequences, such as the exacerbation of individualism and egocentrism in terms of limited resources. The COVID-19 pandemic destabilized society as a “total social fact”, with the population being affected both by the high risk of contracting the virus and by the strict measures of isolation and social distancing [13]. The effect of this health and social crisis has been to exacerbate structural inequalities based on social relations of class, sex, and age [14]. However, there were also positive effects, especially in terms of the community and its role in people’s lives; various actors from the public space mobilized in exemplary fashion to help their peers, in forms in which the norms would not have manifested in normal conditions [15,16]. In Romania, there are numerous studies that analyze the impact of the pandemic on socio-economic aspects [17,18,19,20], on the consumption behavior of the population [21,22,23], on the educational system, and on pupils and students [24,25,26], as well as the way in which the population or its various categories perceived the causes, effects, and management of the pandemic at the local or national level [25,27,28,29,30]. Issues related to the role of social cohesion in managing the effects of the pandemic are addressed either at the national level, compared to the situation in other European countries, in a more general way [6], or tangentially, in studies whose main purpose is different [27,28,31].

In Romania, the beginning of the pandemic was quite slow; in the first month (March 2020) the number of diseases was relatively low (probably due to a very low testing capacity). However, the worrying European context and the strong return migration of those who went to work in countries severely affected by the pandemic at the time (Italy and Spain) led the authorities to impose rather harsh measures from the outset (establishing a state of emergency in the country in March 2020, the transition to online education from 12 March 2020, etc.).

Despite these measures, the situation degenerated in Suceava County, located in northern Romania, with a population of about 700,000 inhabitants, characterized by a strong emigration phenomenon for work. Against the background of faulty management of the County Emergency Hospital from Suceava—the residence of the homonymous county—and of a massive return migration, in the middle of April 2020 this county registered 24.7% of the total number of diseases in Romania and 47.5% of the number of illnesses of medical staff in the country [30]. This situation imposed exceptional measures such as the quarantine of the city and the peri-urban area and the establishment of a military management at the level of Suceava County Hospital (SCH).

Under these conditions, in a climate of fear, concern, and insecurity, and under the background of authorities almost absent, at least at the beginning of the crisis, various local civil society actors, as well as various public and private entities, reacted. In Romania, the tradition of involving civil society in the community is not a very long one. Despite the very large number of officially registered organizations—over 110,000 in 2019, of which only half are truly active—they do not have a real capacity for collective mobilization and operate rather insularly, targeting various vulnerable groups [32]. In fact, studies on the role of NGOs and civil society in Romania [6,32] and in post-communist states [33,34,35] indicate a lack of trust and involvement of citizens in collective action and a low level of solidarity and civic spirit [36] as a result of the totalitarian experience that these states have gone through relatively recently. Changing mentalities takes time, being a consequence of material and cultural accumulations that can only be achieved by changing generations [6].

Civil society has made its presence felt in the Romanian space since 1990, and its forms of manifestation have evolved from those related to the political environment (some NGOs, or their leaders, which over time have become important actors in political life), to those that oversee the functioning of democracy in all its components and, more recently, to actions of a social nature, which come to replace the role of the state where it does not perform [32,37] Among the most visible actors of the civil society at the national level are those concerned with correcting the shortcomings in the Romanian health system, and the campaigns carried out for this purpose enjoy the greatest support and visibility. *MagiCAMP* and *Dăruiește Viața* are only two such examples [38]. At the local level, however, social activism is almost non-existent. In fact, most of the manifestations of the civil society, both at national and local level, took place within campaigns initiated by the media, the church, or various personalities with visibility in the public space; either they were determined by the desire to help in case of natural disasters (floods), tragic events (the Collective fire), or they started from the support/contestation of some ideas or projects (Roșia Montană, Ordinance 13), in which case they were quickly confiscated politically. These expressions of solidarity rarely had a spontaneous character, being in fact impelled by something/someone. Another characteristic of the involvement of the civil society, formal and informal, in the Romanian space is the more active involvement “against something/someone” than “for something/someone”. With very few exceptions, the voice of civil society in the Romanian public space was heard to challenge, not to support.

In this context, it was considered that Suceava County is an adequate space for the analysis of how the difficult situation faced by the population at the beginning of the pandemic influenced the collective involvement (through NGOs and other public and/or private actors) and/or its individuality in solving community problems. The present study aims to answer a few research questions to finally outline a picture of resilience at the collective and individual level, an experiment or test of civic solidarity. What were the ways of involving the population and what is the profile of those involved? How did the population perceive the reaction of civil society in a time of major crisis? To what extent has the involvement of the population been influenced by the way in which the media has reflected this crisis?

In order to decipher the mechanisms of civic involvement/non-involvement, the following hypotheses were taken into consideration:

**H1.** 
*The population has seen the crisis resolved through the involvement of the authorities and passive personal involvement, rather than their own or active group involvement.*


**H2.** 
*The reaction (degree of involvement) at individual and/or collective level was conditioned by the distance from the disease and by socio-demographic factors (age and gender).*


**H3.** 
*The media acted as a catalyst for individual and/or collective involvement.*


## 2. Materials and Methods

The present sociological diagnosis used data that were collected through an online survey at the beginning of May 2020 among the population of Suceava County.

The pandemic period meant a limitation of social interactions, the situation in which the most appropriate way to collect research data was online collection through Google Forms. In order to increase the level of heterogeneity of the sample and to include in the sample as high a diversity of respondents as possible, we resorted to the construction of the sample by snowball method, in this case all participants being asked in turn to send the link of the questionnaire to other people in order to increase the number of participants. The data collection tool was designed as a sociological questionnaire that brought together a total of 33 questions, most of which were predefined questions.

In the beginning, the main limitation was that females were prevalent in the sample, which is why we used a weighting of the sample by gender variable. In this way, we balanced the structure of the sample so that its sample can reflect the characteristics of the reference population (Suceava county population). Another limitation of the study was the unequal access to digital technology, which explains the lower share of respondents in rural areas and those with lower education.

The data collected focused on several aspects, including the perception of the population on the causes underlying the outbreak of COVID-19, but this article focused only on how the intervention of civil society was perceived, the influence which was expressed by the media, and, last but not least, the way in which the participants in the study were involved in resolving the crisis.

The aim was also to identify how the reaction of civil society was perceived and how it was mobilized, as well as how the mass media contributed to reducing the effects of the pandemic. A distinct objective was to measure the level of involvement of study participants and the forms in which they were involved.

After the elimination phase of non-compliant responses, the volume of the sample included a total of 1231 people, of which 67.3% were female and 32.7% were male (the results of the study required a weighting to gender variable). The mean age of the study participants was 38.05 years.

## 3. Results

### 3.1. Civil Society Involvement

At the end of March 2020, in the conditions of the sustained increase in the number of cases of infection with the new coronavirus at the national level, the situation seemed out of control at Suceava County Hospital (SCH). A very large number of infected medical staff, dysfunctional drug and food supply circuits for patients, a climate of panic and insecurity that determined even the unjustified absence of staff, completely outdated management, and a lack of minimum measures to protect both staff and uninfected patients all featured amid the explosive increase in numbers of requests from the beneficiaries of medical services. The local authorities under which the SCH was subordinated reacted late, without firmness, and without proving that they had a coherent crisis management plan. The SCH leadership was fired, but the new interim leadership resigned after only a few days, declaring its inability to manage the situation. Under these conditions, the local and national media presented an apocalyptic situation, an image with a strong emotional impact that accentuated the panic and the feeling of abandonment that had already settled, especially among the population of the county seat, Suceava.

In the face of the almost total lack of reaction of the authorities, more or less known actors of the local civil society started to get involved. The most effective and visible involvement was that of the entrepreneur Ștefan Mandachi who, using the notoriety he already enjoyed locally as a result of a previous action, launched the campaign “1 cm of good deeds-all in the front line” in partnership with the Red Cross Suceava branch and the Society of Medical Students from Iași.

This campaign managed to raise over 1.5 million euros in a short time (the campaign started on 17 March 2020 and lasted 40 days), which was the largest amount ever obtained in Romania through a campaign initiated by an individual. The money, obtained both from substantial donations from companies or personalities from various fields and from symbolic donations from community members, was used to equip hospitals in the cities of Suceava County and to provide support to nursing homes. In addition, the initiator of the campaign supported SCH by offering free accommodation in one of its hotels for medical staff (who worked in 2-week shifts to avoid contact with family and other community members), and by distributing free food to medical staff and SCH patients when the operation of the hotel could no longer be managed with its own resources.

This campaign was the largest and most visible, but it was not the only one. Other associations and organizations, as well as many local entrepreneurs, helped both the medical units in the county and institutions for vulnerable people or simply individuals affected by the pandemic. Among the campaigns and actors noted by the respondents, mention should be made of: Rotary Club Suceava Cetate-End COVID now, Rădăuțiul civic-Emergency Fund for Rădăuți, Support Group for Vatra Dornei, Salvamont Vatra Dornei-Together we help Vatra Dornei, and the Humanitarian Foundation 2001 for Romania.

Another very visible and present actor in the critical period that the city of Suceava went through in March–April 2020 was the “Stefan cel Mare” University of Suceava (USV). USV’s involvement had several components and continued even after the critical period was over. Among the most important actions of USV, one can mention the implementation of a complete line of semi-automatic COVID-19 tests, which were transferred to SCH; the creation and implementation of the Counseling Center for both medical staff and students and people emotionally affected by the pandemic through launching a telephone line for psychological, medical, and logistical counseling; and the digitization of the activity of the Public Health Directorate (DSP) Suceava. In addition, USV teachers, researchers, and students volunteered to train and assist both medical staff and those in need.

The Romanian Orthodox Church (B.O.R.) is another essential component of Romanian civil society that enjoys great trust among the population. Together with the national army, BOR ranks first in opinion polls in terms of public confidence in state and non-governmental institutions [39]. The church also acted in support of the Suceava community, getting involved through the direct purchase and donation of medical equipment; by initiating platforms for donations (www.Alăturilagreu.ro, accessed on 7 April 2022) and humanitarian campaigns (Together, we help Suceava), the funds thus obtained being intended for the purchase of medical equipment and hospital supplies; and by organizing teams of volunteers to provide assistance to vulnerable people.

The challenges of the pandemic and the tension existing at that time in Suceava County contributed, as previously shown, to the involvement of civil society and to the mobilization of several actors in the public space. In this regard, study participants were invited to mention some of the initiatives in support of the fight against COVID-19 that they noted in the public space. The inventory of the free answers offered by the respondents allowed us to make a graphic representation in the form of a cloud of words (Figure 1) of the most common answers:

An analysis of the word cloud highlights the following aspects: donations and compliance with the rules were the most visible manifestations of society in the face of the crisis; most of the actions taken were aimed at supporting the medical system and vulnerable groups (the elderly or disadvantaged people); in public perception, Ștefan Mandachi imposed himself as the initiator, leader, and catalyst of the actions undertaken by civil society; and solidarity and involvement appear to be important elements in respondents’ perceptions of the fight against COVID-19. The heterogeneity of the inventoried answers required their grouping and the outlining of initiatives with a higher degree of generality. However, where the category of answers allowed the identification of distinct subcategories, they were mentioned. Almost ¼ of the participants in the study mentioned elements that can be attributed to the activities undertaken by civil society, both in an organized form through the initiative of non-governmental organizations and by the inclusion of initiatives that belonged to individuals. The initiative taken by the entrepreneur Ștefan Mandachi, who initiated the campaign “1 cm of good deeds-all in the first line”, reaching a level of notoriety of 72% of the total of the mentioned initiatives, definitely stood out from the category of non-governmental organizations.

A distinct category of responses was represented by initiatives that directly aimed at supporting the activity of the medical sector (19%). This category of responses included a variety of mentions that respondents listed, including support for physicians and medical staff with protective equipment (masks, gloves, coveralls, and visors), with material and financial resources to enable the purchase of medical equipment, emotional support, and the provision of accommodation for medical staff so that they do not come into contact with family or other persons when they were at rest.

The third category of initiatives that remained in the collective mind concerned the campaigns that were carried out mainly in the media and aimed at respecting the rules of personal hygiene, quarantine, and maintaining social distance in public spaces. While the first two categories of initiatives listed directly concerned the health system and the work of civil society (which also indirectly acted in support of the health system), this category of initiatives focused exclusively on how individuals directly contributed in the fight against COVID-19 through personal actions and behavior.

The epidemiological crisis that Suceava County faced at the beginning of the pandemic period was a challenge that brought together the combined efforts of several actors from different levels of the community organization. In this sense, the inventory of initiatives to combat COVID-19 allowed us to group them according to the level of involvement of the actors. Thus, two categories of initiatives were outlined: institutional initiatives and respectively individual initiatives.

The category of institutional initiatives included the response of various institutions in reducing the spread and limiting the effects of the pandemic. Thus, included in the category of institutional initiatives were the actions of non-governmental organizations (www.1cm.ro (accessed on 7 April 2022) and the Red Cross), but also of public institutions (Suceava City Hall), Ștefan cel Mare University of Suceava, and the church.

The relevance of institutional initiatives (63.3%) at the level of the collective mind proved to be more consistent compared to that of individual initiatives (36.7%). In relation to the environment of residence, the institutional initiatives are mainly associated with the urban environment (χ^2^ (1) = 14.59, *p* = 0.00) and the localities that were quarantined (χ^2^ (1) = 18.052, *p* = 0.00), while the individual initiatives are mainly mentioned by the respondents from the rural area, especially from the non-quarantined localities (Table 1).

### 3.2. Personal Involvement in the Fight against COVID-19

The analysis of how the involvement of civil society in the fight against COVID-19 was perceived requires the consideration of the way in which the participants in the study were involved. The category of individual initiatives included those that came from individuals (volunteering and donations for a specific purpose) aimed at internalizing the rules of maintaining personal hygiene (washing hands with soap for a few minutes), adopting behaviors that limit the transmission of the virus (wearing protective masks), and supporting disadvantaged people (by providing medical equipment, masks, and food) and sick or destitute people at home (by supporting the purchase of medicines, food, or basic necessities, etc.).

The analysis of the answers showed that 81% of the total participants in the study were involved in the fight against COVID-19 through at least one of the following forms: by volunteering, by making donations (financial, food materials, or food), or by helping vulnerable people. Alternatively, 19% did not show solidarity with the challenges of the pandemic Suceava County faced.

From a structural point of view, the individual involvement was differentiated (percentage of the total of those who stated that they were personally involved). 

The analysis of the results shows a relatively low level of individual involvement through voluntary actions (approx. 14%), the considerable share being limited to supporting the activities of civil society, either through donations (40%) or by supporting vulnerable people in extreme situations (46%).

An important element that can contribute to shaping an overview of personal involvement in mitigating the effects of the pandemic may be personal experience in relation to the actual illness. Thus, starting from the answers provided by the study participants, two categories of respondents were outlined: the category of respondents who were diagnosed with COVID-19 themselves or had family members who were infected (subjective experience), and the category of respondents who had social proximity to people who were ill or even died (contextual experience).

The involvement of people who had contextual experiences in relation to the activities in which they were actually involved, highlights the predominance of help to vulnerable people, donations, and, last but not least, involvement through volunteering (Table 2). On the other hand, those who had subjective experiences in relation to the disease with COVID-19 were mainly involved in activities to support vulnerable people (χ^2^ (1) = 5.17, *p* = 0.02).

In order to have a more complete picture of the profile of the people who were involved in the activities to reduce the effects of the pandemic, we collected series of variables of a sociodemographic character (age and gender), as well as variables that will highlight the personal experience of the respondents in relation to COVID-19. Thus, it can be remarked that the average age of the people involved (Table 3) in the activities of mitigating the effects of the pandemic (38.6) differs statistically significantly from the average age of those who did not make any effort in this regard (average age of people involved: 38.6 years, std. dev. = 11.6; average age of people uninvolved: 35.6, std. dev. = 13, t (324.29) = 3.24, *p* = 0.00). The average age of the people involved is 3 years older than that of those who did not join the fight to reduce the effects of the pandemic. This difference highlights the fact that those involved understood the need to support this approach and the difficulties faced at that time by the community in Suceava County. Last but not least, there was the possibility of personal fears, especially as in the public space at that time there circulated the information that the pandemic can generate negative effects among older people.

Referring to the concrete way in which the study participants were involved, for all three forms of involvement the average age of those involved was statistically differentiated from the age of those who did not make efforts, having slightly higher values, as follows: involvement through volunteering (average age of those involved: 39.4 years, std. dev. = 10.6, average age of those not involved: 37.7, std. dev. = 12.2, t (349.18) = −2.06, *p* = 0.00); involvement by making donations (average age of those involved: 38.9 years, std. dev. = 10.9, average age of those not involved: 37.1, std. dev. = 12.9, t (1105.18) = −2.55, *p* = 0.01); and the involvement through help offered to vulnerable people (average age of those involved: 38.7 years, std. dev. = 11.5, average age of those not involved: 36.8, std. dev. = 12.6, t (881.32) = −2.60, *p* = 0.00).

Regarding the involvement of the study participants in the fight against COVID-19 in relation to gender (Table 4), a predominantly female participation can be observed, both in terms of overall involvement (χ^2^ (1) = 18.44, *p* = 0.00) as well as involvement in donation support activities and support for vulnerable people.

All these manifestations of formal civil society have been visible in the local community. The perception of the participants in the study on the way in which the civil society was involved in diminishing the effects of the pandemic during the beginning of the crisis has a predominantly positive connotation, in the context in which approximately three quarters of the respondents (72%) appreciate the interventions of the civil society being very good and efficient; alternatively, 28% of the respondents considered the interventions of the civil society rather modest. As expected, assessments of civil society activity may be associated with or determined by a number of factors that have acted latently in the context of the pandemic. With regard to informal civil society, that is ordinary citizens, most respondents stated that they were involved in helping vulnerable people; fewer mentioned donations as a form of involvement, and fewer still said they were involved through volunteer work.

A first element that marks the perception of civil society interventions is the age of the study participants. Thus, we can observe the existence of statistically significant differences in terms of the age of the respondents who expressed themselves regarding the intervention of the non-governmental environment in the fight against COVID-19 (Table 5). We find that the positive assessments come from people with a higher average age (39.9), compared to the more reserved assessments of younger respondents (33.9).

The rather passive and impersonal involvement of respondents in resolving the Suceava crisis, as well as the perception of the same type of action as important in limiting the effects of the pandemic, indicates a certain stage in the evolution of local civil society after 30 years of democracy: people still expect solutions from the outside, but there are still germs of a proactive attitude, through which the individual and society can become relevant actors in crisis management.

Analyzing the profile of those involved in the fight against COVID-19 in relation to the distance from the disease, we can see the trend of higher involvement from people who were in social proximity with people who got sick or even died compared to people who did not have this unfortunate experience (Table 6). In other words, when there were certainties (confirmed by people through social proximity: more distant relatives, friends, acquaintances, etc.) regarding the results of contamination, people mobilized and made an effort in this regard. One can also notice a slight association between involvement in the fight against COVID-19 and subjective experience, but this time the association is at the limit of statistical significance. However, one can appreciate that the relatively limited involvement of those who had subjective experiences can be explained either by the fear of relapse or by their limited ability to get directly involved in this struggle.

### 3.3. Mass-Media (Local)—Catalyst in the Involvement of the Population in the Fight against COVID-19

An important factor in empowering the population in the fight against COVID-19 was the way in which the reflection of the situation, created in the local and national media, was perceived. As expected, the participants in the study appreciated that the local media presented an image closer to the truth compared to the national media. These beliefs could also be due to the fact that the local media reflected realities from the geographical proximity and could have known and understood the local context in more depth, but also due to the fact that the participants in the study showed a high level of trust in the media.

Thus, it was found that the positive assessments related to the way in which the local media reflected the situation of the pandemic in the county are associated with the positive assessments of the national media, χ^2^ (1) = 242.25, *p* = 0.00

An important element that contributed to the positive appreciation of the way in which the local media reflected the state of the pandemic can also be explained by the existence of contextual experiences arising from the illness of people in social proximity or even the death of acquaintances (Table 7), χ^2^ (1) = 10,543, *p* = 0.00. Thus, when there were such contextual experiences, the appreciation of the activity of the local media was rather positive, compared to the situations in which the respondents did not have victims of the pandemic in their social proximity. This confirms that the level of credibility of how reality has been reflected through local communication channels is also strengthened as a result of personal experiences.

The involvement of study participants in the fight against COVID-19 materialized through various actions and was stimulated at the individual level by the way in which the local media reflected the reality of that period.

Thus, it has been considered that the association between offering donations (in money, products, food, etc.) and the appreciation of the local media as objective and close to the truth is not accidental (Table 8) (χ^2^ (1) = 22.858, *p* = 0.00), the media directly contributing to the population mobilization in the process of collecting resources in the form of donations.

## 4. Discussion

According to the obtained results, it is obvious that, in the situation of the lock-down in Suceava and the neighboring communes, the civil society organizations took over tasks that the state (central and local authorities) should have managed. The security of individuals and the community became, in whole or in part, the responsibility of some civil society organizations through their responsible involvement. The involvement of civil society in solving the problems associated with the pandemic is a common fact all over the globe [16], which differs in the degree and manner of involvement. The results of studies analyzing the role of social cohesion and civil society in managing the crisis caused by the COVID-19 pandemic in European countries [40,41,42,43,44,45] or outside the European space [3,5,46,47] indicate that the success of managing a crisis situation depends on the reaction of civil society, even if the authorities react very well. The cited studies also show that the involvement of civil society can be very different depending on the moment, the context, and the cultural environment. The existence of differences in the involvement and role of civil society between eastern and western European states has also been demonstrated [48].

If, in western societies with an active and present civil society in normal times, the pandemic context only generated an adaptation of support actions to new needs [49], in post-communist societies the current situation sometimes required awakening a civil society that many did not even know existed. Additionally, the mobilization of the Suceava civil society can be considered illustrative for the Romanian society, with a reduced seniority in democracy compared to the countries with a long democratic experience.

Comparing the results of our study with those of the aforementioned studies on the role of civil society in managing the COVID-19 crisis, we find that, at least in Suceava, the health crisis has “awakened” civil society, the effects of its actions being beneficial. In the short term, civil society actions have strengthened social cohesion and had only positive effects. At the same time, the actions of civil society in Suceava have not been very diverse, nor very creative, being limited to classic support through donations and possibly volunteering. Through these characteristics, the actions of the civil society in Suceava are integrated in the model associated with the former communist states, with a limited experience regarding the involvement of the civil society in solving the problems of the community.

The involvement of citizens in limiting the effects of the pandemic has been significant. This aspect is very different from the results of a sociological survey conducted in 1998, at the request of the Foundation for the Development of Civil Society in Romania, which revealed that the reason why Romanians joined a non-governmental organization (NGO) was to obtain aid (22% of the total respondents), the concern for defending group interests being secondary (14%) [50].

Prior to the pandemic, in Suceava there were sporadic demonstrations of solidarity in humanitarian campaigns organized by local institutions or individuals, generally to help people in need from various points of view. It can be stated, however, that in accordance with the situation at regional and national level, the actions of civil society in normal conditions in Suceava are still timid, and the civic spirit is maturing. The reaction of the Suceava community in the face of a major crisis indicates a tendency to crystallize the role of civil society. It can be deduced that, in the context of the pandemic, civil society presents, as an essential trend, the return to normal, in the sense of consciously assuming its role and functions in a democratic society. However, the way in which the civil society of Suceava reacted and mobilized in the face of the crisis indicates a change of attitude, a step forward in terms of its transformation from a marginal, insignificant actor to a participatory actor, whose actions can generate change.

The media, as a component of civil society, managed to mobilize important segments of the population, both in quarantined localities and in other localities in Suceava County and Moldova. At the same time, the civic actions of the media highlight the role assumed not only to inform its citizens, but also its concrete commitment in the fight against COVID-19 and the promotion of a “civic culture”. The local media made a radical contribution to reducing public apathy and shifting personal and public concerns. Social networks were used extensively, and the easy and efficient communication of the people present on these platforms allowed an unprecedented mobilization: from the emergence of new solidarity practices consisting of multiplication, collection, and distribution of food; planning and organizing actions; to crowdfunding (raising funds by selling 1 cm of highway).

Age, place of residence, and distance from the disease are the variables that most influenced the perception of the population on the involvement of civil society in resolving the crisis. The fact that a more favorable perception of how civil society reacted was more characteristic of people with a higher average age can also be explained by the informational context of that period when it was mentioned that older people are more at risk. In this context, older people were tempted to consider positive almost any initiative that would have contributed to reducing the effects of the pandemic. The greater visibility of institutional initiatives for urban respondents is also explained by both the higher level of information and the more rigorously imposed and practiced distance in urban areas, especially at the height of the crisis, which made personal, direct involvement more difficult.

## 5. Conclusions

The COVID-19 crisis tested social cohesion and the resilience of communities both locally and nationally. Despite the perception that Romanians would rather be part of a culture of individualism, in extreme situations it was found that their activity has never been more important. The mitigation of the negative impact of the Suceava crisis also largely depended on the ability of civil society to strengthen its role through its response and effective actions. The analysis of the activity of the different components of the civil society from Suceava County indicates a slight, but certain, tendency of the maturation of the civil society and of the diminution of the *contesting character*, of confronting the civil society with the power. Thus, more and more insistently, the civil society was involved, through its own specific ways, in promoting the general interests of communities. Active involvement in the fight against the pandemic indicates a certain stage in the evolution of local civil society after 30 years of democracy. Although, in many cases, external solutions are still awaited, there are still signs f a proactive attitude, through which the individual and society can become relevant actors in crisis management. The reaction and actions of the Suceava community in the face of a major crisis demonstrate a tendency to crystallize the role of civil society and return to normal, in the sense of consciously assuming its role and functions in a democratic society. This partially confirms the first hypothesis that underpinned this study.

Regarding the profile of the persons involved in diminishing the effects generated by the pandemic crisis, it is to be appreciated that, in general, there was a more consistent involvement of females, that the persons involved were more mature in terms of age compared to those who were not involved, and contextual experiences and the existence of the disease in the vicinity generated a better mobilization. Thus, the study indicates that the reaction at the individual or group level was nuanced by various variables, thus confirming the second research hypothesis. An important role in empowering the population to reduce the effects of the pandemic was also played by the way in which the media managed to present the reality of that period. The public’s trust in the activity of the local media and the favorable appreciations towards them (the media were objective and correctly reflected the reality) were associated with the involvement of the population in this fight. This confirmed the catalytic role that the local media played in mobilizing civil society.

Although the survey involves a number of limitations due to the structure of the sample (gender, access to technology, or studies), the results of this study suggest the existence of a potential for action at the level of Romanian civil society, at least in exceptional conditions, and the reaction of Romanian civil society in the current crisis of Ukrainian refugees (in which Suceava County again plays a major role, being a border county with Ukraine) proves this. In this context, the analysis of the role of civil society in managing a crisis (in this case, generated by a pandemic) in Suceava may be a starting point for further studies that will analyze other manifestations of civil society in Romania in more or less similar conditions. Civil society played a key role, through its actions at all levels, offering a remarkable lesson in post-December solidarity that reflects the idea of active, united, and inclusive citizenship. 

## Figures and Tables

**Figure 1 ijerph-19-04587-f001:**
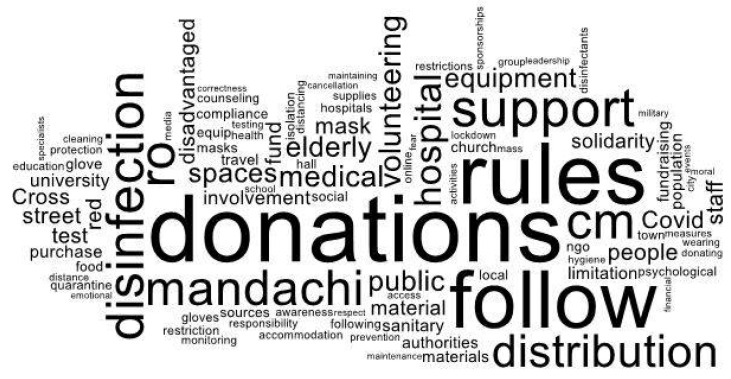
Initiatives of civil society and public actors in the fight against COVID-19 (Source: Survey, 2020).

**Table 1 ijerph-19-04587-t001:** NGO initiatives/residence environment and status of locality.

Variables	N	Initiatives		χ^2^	df	*p*
		Individual	Institutional			
Environment of residence				14.59	1	0.000
Urban	746	249	497			
Rural	245	115	130			
Quarantined locality				18.052	1	0.000
Yes	534	164	370			
No	457	200	257			

**Table 2 ijerph-19-04587-t002:** Table of associations: Personal involvement in the fight against COVID-19—contextual experience/distance from the disease.

Variables	N	Contextual Experience		χ^2^	df	*p*
		No	Yes			
Involvement through volunteering				6.625	1	0.01
No	944	441	503			
Yes	211	78	133			
Involvement through donations				17.846	1	0.00
No	546	281	265			
Yes	609	238	371			
Involvement through supporting the vulnerable people				30.757	1	0.00
No	439	243	196			
Yes	717	277	440			

**Table 3 ijerph-19-04587-t003:** Personal implication/age of participants.

Variables	n	Mean	Std. Dev.	t	df	*p*
Involved in The Fight Against COVID-19				3.24	324.29	0.001
Uninvolved	233	35.61	13.005			
Involved	998	38.62	11.643			

**Table 4 ijerph-19-04587-t004:** Personal implication/gender.

Variables	N	Gender		χ^2^	df	*p*
		Female	Male			
Involvement by donations				18.523	1	0.00
No	587	258	329			
Yes	640	360	280			
Involvement through help offered to the vulnerable people				11.911	1	0.00
No	473	209	264			
Yes	755	410	345			

**Table 5 ijerph-19-04587-t005:** The reaction of civil society/age of participants.

Variables	n	Mean	Std. Dev.	t	df	*p*
The reaction of civil society				7.62	534.68	0.000
Pretty modest	318	33.93	12.274			
Very good and efficient	822	39.98	11.248			

**Table 6 ijerph-19-04587-t006:** Table of associations: Personal involvement in the fight against COVID-19/distance from the disease.

Variables	N	Personal Involvement in The Fight against COVID-19		χ^2^	df	*p*
		Uninvolved	Involved			
Subjective experience				4.041	1	0.04
No	1095	236	859			
Yes	88	11	77			
Contextual experience				28.753		0.00
No	519	142	377			
Yes	637	93	544			

**Table 7 ijerph-19-04587-t007:** Table of associations: Personal involvement in the fight against COVID-19—contextual experience/appreciation of the local media.

Variables	N	Contextual Experience		χ^2^	df	*p*
		No	Yes			
Local media				10.543	1	0.00
Exaggerated, even biased	293	150	143			
Objective, close to the truth	712	285	427			

**Table 8 ijerph-19-04587-t008:** Table of associations: reflecting the situation in the local media and personal involvement through donations.

Variables	N	Through Donations		χ^2^	df	*p*
		No	Yes			
Local media				22.858	1	.00
Exaggerated, even biased	307	178	129			
Objective, close to the truth	753	315	438			

## Data Availability

The data presented in this study are available upon request from the appropriate author. The data is not available to the general public. The approved public, with competences in the analysis and interpretation of data from sociological studies will be able to request access to the study database.

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
