# Peer review of "Social Cohesion and Community Resilience during the COVID-19 Pandemic in Northern Romania"

_ijerph, 2022, doi:10.3390/ijerph19084587_

Round 1

Reviewer 1 Report

 Social Cohesion and Community Resilience during the Covid  19 Pandemic in Norhem  Romania

This is an interesting and timely topic.  The authors have developed a relatively large nonprobability sample to examine how social cohesion plays out in the context of the COVID-19 pandemic response.  The research has potential but a few things need to be added and refined.

  1. The article would be strengthened by adding a set of research questions early on in the paper. You wait to tell readers what the paper is about. Tell them upfront. It helps in making your argument easier to follow.
  2. In the methodology section, you talk about how you did the survey and how you selected the subjects. You have a large accidental or convenience sample that you collected with a snowball sampling procedure. Why did you do it that way?  What are the limitations of the procedure?  Are the demographic statistics for the local population consistent with those of your sample?
  3. The data collection technique rules out those without access to technology (the digital divide). How does that affect your findings?
  4. On page 5, please explain the word cloud a bit more. I would give real thought to replacing the Pie Chart with something else.  At any rate, it needs more explanation.
  5. Is notoriety really the word you want on Line 255?
  6. Table 3 on page 8 needs to be better developed and the discussion needs to be clearer (also true of the tables after that). Students T is a parametric statistic. Would you be better off with a nonparametric statistic, such as the Mann-Whitney U?  I  think you can make an argument either way.
  7. On page 9, you need a transition paragraph after the table.
  8. Tables 7 and 8 the discussion about it needs to go in the results section. This is not where you introduce new material.
  9. On page 11, is the Media really a component of civil society?
  10. There needs to be a discussion of the limitations of the study in the conclusions

Good luck with the article

Author Response

Dear Reviewer,

We modified the manuscript according to observations / remarks received. We accomplished the requirements. Please find below the answers to your observations.

Thank you for all the comments/ remarks you have made, and we hope you will agree with the publication of this article!

Despina Saghin and the co-authors

Reviewer 2 Report

The research topic presented in the manuscript is dealing with a contemporary and important topic. It is a well-written manuscript; however, I suggest to authors include these points:
1.    This study should clearly show the knowledge gaps identified and link them to your paper goals.
2.    Please reason both the novelty and the relevance of your paper goals.
3.    In the literature, articles that contribute significantly to the topic of the study can be identified. By querying the Web of Science database and the MDPI site, you can identify articles that can be cited to improve this section.
4.    To improve the scientific soundness of the study, it would be interesting to check whether there are studies on social cohesion and community resilience in other regions of Romania / Europe.
5.    Follow the journal-specific article formatting requirements. Observe the requirements for citing bibliographic sources in the text. The tables do not appear to follow the log format. Please check and fix it.
6.    English language and style are fine/minor spell check required. Please ask for professional service for an English check. Also, the paragraphs beginning with lines 46 and 229 are not clear.
7.    Avoid using the personal pronoun (we, our).
8.    Present in the conclusions the innovative character of the research carried out and future research perspectives.

Author Response

(The authors gave the same response as above.)

Reviewer 3 Report

This paper developed an empirical study using data collected from Suceava County in Norhem Romania to investigate how the civil society reacts to the Covid 19 Pandemic, how the inhabitants perceives the involvement of civil society, and what is the role of the mass media during the pandemic. The study does fall within the scope of the journal. The story line is basically clear, and the results of the quantitative analysis are interesting. In order to further improve the quality of this paper, I have several suggestions.

  1. According to the title, the main theme of this study is “community resilience”. However, the results of the paper are centered on how civil society, individuals, and the mass-media are involved to fight against Covid 19, instead of how they react to enhance community resilience. You may need to define how you measure “resilience” and emphasize the keyword in the Results and other sections.
  2. Further strengthen the comparison between the results of this paper and similar studies in other countries, so as to form a more worldwide and systematic understanding of the role played by the civil society during Covid 19.
  3. What are the implications from findings of this study?

Small issues:

  1. Some of the parentheses are incorrectly used, e.g., Line 49, 90, 275.
  2. Line 73, what does “SCH” mean?
  3. Line 128, change “I” to “We”.

Author Response

(The authors gave the same response as above.)

Round 2

Reviewer 1 Report

Changes made to the revised manuscript (with track and changes)

Please find below all changes made in the corrected manuscript with track and changes in correlation with the lines at which they appear:

  1. The article would be strengthened by adding a set of research questions early on in the paper. You wait to tell readers what the paper is about. Tell them upfront. It helps in making your argument easier to follow.

The present study aims to answer a few questions to finally outline a picture of resilience at the collective and individual level, an experiment or test of civic solidarity. What were the ways of involving the population and what is the profile of those involved? How did the population perceive the reaction of civil society in a time of major crisis? To what extent has the involvement of the population been influenced by the way in which the media has reflected this crisis?

This is what I wanted. Please label this as Research Questions.  These should also be mentioned in the discussion/conclusion

  1. In the methodology section, you talk about how you did the survey and how you selected the subjects. You have a large accidental or convenience sample that you collected with a snowball sampling procedure. Why did you do it that way?  What are the limitations of the procedure?  Are the demographic statistics for the local population consistent with those of your sample?

The pandemic period meant a limitation of social interactions, the situation in which the most appropriate way to collect research data was online collection through Google Forms. In order to increase the level of heterogeneity of the sample and to include in the sample a diversity of respondents as high as possible, we resorted to the construction of the sample by the method of the snowball.

This is what I wanted but I cannot find it in the paper. This probably should be on page three.  You use “I” but there are multiple authors—is this a good idea.?

  1. The data collection technique rules out those without access to technology (the digital divide). How does that affect your findings?

In this way, the selection of the respondents was made randomly, each of the participants being invited to distribute the questionnaire to as large a number of people as possible. In the beginning, the main limitation was that there were several females in the sample, which is why we used a weighting of the sample by gender variable. In this way we balanced the structure of the sample so that its sample can reflect the characteristics of the reference population (Suceava county population).

You can’t say its random if you used Chain referral or “snowball sampling”.  How about adding the digital divide (access to technology” as a limitation of the study?  Weighing (as you say) is the proper response.

  1. On page 5, please explain the word cloud a bit more. I would give real thought to replacing the Pie Chart with something else.  At any rate, it needs more explanation.

The analysis of the word cloud highlights the following aspects: donations and compliance with the rules were the most visible manifestations of society in the face of the crisis; most of the actions taken were aimed at supporting the medical system and vulnerable groups (the elderly, disadvantaged people); in the public perception, Ștefan Mandachi imposed himself as the initiator, leader and catalyst of the actions undertaken by the civil society; solidarity and involvement appear to be important elements in respondents' perceptions of the fight against Covid 19.

We gave up the Pie Chart.

Looks good!  Thank you for getting rid of the pie chart

  1. Is notoriety really the word you want on Line 255?

We replaced the word ”notoriety” with the word ”relevance”.

Excellent!

  1. Table 3 on page 8 needs to be better developed and the discussion needs to be clearer (also true of the tables after that). Students T is a parametric statistic. Would you be better off with a nonparametric statistic, such as the Mann-Whitney U?  I think you can make an argument either way.

The interpretation associated with Table 3 shows that there are statistically significant differences between the ages of those involved and those not involved in the fight to limit the effects of COVID 19. Also, statistically significant differences are maintained when the analysis is performed on those three types of involvement: involvement through volunteering, involvement by making donations and the involvement through help offered to vulnerable people.

Regarding the quality of the variables and the option for the t test on independent samples, we opted to use this test because the independent variable used is a parametric variable (age), respectively dependent variable is a dichotomous variable measured on a nominal scale (level of involvement).

We considered the option of using non-parametric statistics elements (Mann-Whitney U), but we appreciated that the use of the student t test is also appropriate.

That’s fine.  I was just raising the issue.  What you did is completely appropriate

  1. On page 9, you need a transition paragraph after the table.

The text has been restructured as recommended

Very Nice!

  1. Tables 7 and 8 the discussion about it needs to go in the results section. This is not where you introduce new material.

We restructured the Results and Discussions sections, introducing tables 7 and 8 and their comments in the Results section.

Good.  Well done

  1. On page 11, is the Media really a component of civil society?

There are authors who consider that the Romanian civil society includes: non-governmental organizations, trade unions and the media (Dobrescu, 2000). Other authors include trade unions, churches and non-governmental organizations in civil society, as well as the family (Muresan and Dutu, 2006). It can be said that civil society is made up of: the media, trade unions, churches, non-governmental organizations and the family. 
Thus, an important component of civil society is represented by the media, of course the independent one, not the one subordinated to the interests of a political party or a coalition of parties

Fair Enough.  You might want to note that this is different from the conception in the US and UK (and totally reasonable) where the media is part of the corporate world.

  1. There needs to be a discussion of the limitations of the study in the conclusions

The limitations of the study, related to the methodology, were specified in the Materials and methods section.

 I’m sorry but I can’t find that.  There are limits to the research design, data collection and sampling strategy (as in all research) but I cannot find where it was discussed.  Having this in here makes it less likely that you will be criticized by other scholars

One additional issue

You state:

The results of this study prove the existence of a potential for action…

Really it doesn’t prove that.  It suggests that. You’ll have no end of criticism if you say it like that.

Author Response

(The authors gave the same response as above.)

Reviewer 2 Report

Dear Authors, 

Thank you for taking the suggestions into account. The study is improved and clearer. As a result, I am satisfied that the documents meet the published criteria.

Author Response

Dear Reviewer,

Thank you for your consent to the publication of this article!

Despina Saghin and the co-authors
